# Tracking Reservoirs of Antimicrobial Resistance Genes in a Complex Microbial Community Using Metagenomic Hi-C: The Case of Bovine Digital Dermatitis

**DOI:** 10.3390/antibiotics10020221

**Published:** 2021-02-23

**Authors:** Ashenafi F. Beyi, Alan Hassall, Gregory J. Phillips, Paul J. Plummer

**Affiliations:** 1Veterinary Microbiology and Preventive Medicine, Iowa State University, Ames, IA 50011, USA; afbeyi@iastate.edu (A.F.B.); gregory@iastate.edu (G.J.P.); 2Veterinary Diagnostic and Production Animal Medicine, Iowa State University, Ames, IA 50011, USA; ahassall@iastate.edu; 3National Institute of Antimicrobial Resistance Research and Education, Ames, IA 50010, USA

**Keywords:** antimicrobial resistance genes, digital dermatitis, Hi-C ProxiMeta, resistomes, tetracycline resistance

## Abstract

Bovine digital dermatitis (DD) is a contagious infectious cause of lameness in cattle with unknown definitive etiologies. Many of the bacterial species detected in metagenomic analyses of DD lesions are difficult to culture, and their antimicrobial resistance status is largely unknown. Recently, a novel proximity ligation-guided metagenomic approach (Hi-C ProxiMeta) has been used to identify bacterial reservoirs of antimicrobial resistance genes (ARGs) directly from microbial communities, without the need to culture individual bacteria. The objective of this study was to track tetracycline resistance determinants in bacteria involved in DD pathogenesis using Hi-C. A pooled sample of macerated tissues from clinical DD lesions was used for this purpose. Metagenome deconvolution using ProxiMeta resulted in the creation of 40 metagenome-assembled genomes with ≥80% complete genomes, classified into five phyla. Further, 1959 tetracycline resistance genes and ARGs conferring resistance to aminoglycoside, beta-lactams, sulfonamide, phenicol, lincosamide, and erythromycin were identified along with their bacterial hosts. In conclusion, the widespread distribution of genes conferring resistance against tetracycline and other antimicrobials in bacteria of DD lesions is reported for the first time. Use of proximity ligation to identify microorganisms hosting specific ARGs holds promise for tracking ARGs transmission in complex microbial communities.

## 1. Introduction

Bovine digital dermatitis (DD) is a contagious polymicrobial complex disease that causes lameness and is the second most significant health problem, after mastitis, on dairy farms. Its effects include reduction of milk production, loss of fertility, premature culling, deterioration of animal welfare, and increased treatment expenses [1,2,3]. DD is widely distributed across the world in dairy cattle, but it has also been recognized in beef cattle, sheep, and other ruminants [4,5,6].

High prevalence rates of DD have been reported from various countries. For instance, 34% in Canadian dairy cattle [7]; 38% to 41% in French dairy cows [2]; 21.2% in Dutch dairy cattle [8]; 32.2% in dairy cows and 10.8% in beef cows of Victoria, Australia [9]; and 0.5% to 21% in beef cattle in the United Kingdom [4]. Prevalence rates ranging from 31% to 89% (with monthly within-herd morbidity rates ranging from 0.5% to 12%) were reported in southern California dairy herds in the 1990s [10]. Furthermore, DD can occur in outbreaks on dairy farms. In South Africa, 72% of lactating cows in a dairy herd were affected with recurrence after seven months, infecting 37% of the lactating herd, of which 48% represented new cases [11]. Moreover, DD results in substantial direct and indirect economic losses on dairy farms. Relun and colleagues [2] conducted a six-month follow-up study on French dairy farms, wherein they reported 0.50 to 0.70 kg/day less milk yield in a DD cow compared to the healthy cow. The predicted average cost of lameness per individual cow attributed to DD was 133 USD per case [12].

DD is a multifactorial disease of cattle. Factors related to the environment, management, and individual animals play a combined role in its pathogenesis, with the leading players being infectious agents [13]. The exact etiologic agents of DD are yet to be determined; however, several bacterial species have been detected in the lesions [5,14,15], which earned it the name “polymicrobial infectious disease” [5]. While there is insufficient evidence to suggest the involvement of viruses and fungi in its pathogenesis, bacteria are substantiated as the leading infectious causes of this disease on the basis of the effectiveness of antibiotic treatment and the detection of bacteria in the DD lesions [5,16,17].

While different bacterial species are implicated in DD, critical microorganisms include Spirochaetes, particularly the genera *Treponema*, *Mycoplasma*, *Fusobacterium*, *Porphyromonas*, *Bacteroides*, *Campylobacter*, and *Borrelia*, as well as *Dichelobacter nodosus* and *Candidatus Aemobophilus asiaticus* [15,18,19]. Spirochetes, such as *Treponema medium*, *Treponema denticola*, and *Treponema phagedenis*, are predominant in active lesions compared to other bacteria [14,20,21,22,23,24,25,26]. Moreover, studies show that the types and abundances of bacteria involved vary with the stage of the lesions. This suggests that various bacterial species have distinct roles at different stages of the disease, as well as reinforcing the polybacterial nature of DD [14,26,27]. Cultivation of many of these bacterial species in vitro remains a significant challenge.

Control strategies for DD include local or systemic antibiotic or antiseptic solutions for clinical cases, surgical removal of horny proliferation in the case of highly proliferative lesions, foot baths that contain formalin or heavy metals such as copper sulfate and zinc sulfate, and good hygienic and biosecurity measures on farms [13]. Topical applications of oxytetracycline on lesions are commonly used to treat individual cows suffering from DD. This treatment has been reported to yield positive responses [11,28,29]. However, anecdotal reports of clinical resistance exist that suggest the emergence of antimicrobial resistance among causative agents of DD [10,30,31].

Tetracyclines are relatively safe and low-cost drugs that are widely used broad-spectrum antimicrobial agents with activity against a broad range of bacteria, including anaerobic and aerobic Gram-positive and Gram-negative species, along with cell wall-free mycoplasmas and other bacteria [32]. Resistance to tetracyclines continues to rise globally, which was first reported in 1953 in *Shigella dysenteriae* isolates [32]. Recently, dairy producers have been challenged by restrictions on the use of medically important antibiotics in food-producing animals, which are aimed at mitigating the development of antimicrobial resistance in humans [33], as well as recurrence of DD cases. *T. phagedenis* isolates from cows with DD lesions have demonstrated intermediate minimum inhibitory concentration (MIC) values for oxytetracycline [31,34]. In a previous study, antimicrobial resistance genes (ARGs) conferring resistance to copper and zinc were reported in DD cases, but oxytetracycline resistance determinants were not detected [19]. To our knowledge, there are no published works on tetracycline resistance in DD except for the MIC study conducted by Yano et al. [31], and several of the available methods are incapable of identifying ARGs in non-cultivable etiologic agents.

Cultivation-independent metagenomics of environmental samples, such as whole-genome shotgun sequencing, has been used to discover ARGs in complex environments [35]. However, the method is incapable of definitively identifying the bacterial reservoirs associated with specific ARGs. Proximity ligation methods such as Hi-C and 3C have been developed to detect interactions between DNA molecules originating in the same cell, including within microbial communities [36,37,38,39]. As a result, these methods can reconstruct metagenome-assembled genomes (MAGs) from bacterial communities such as those of the mammalian gut communities [37,40]. This method has also been used to identify bacterial taxa hosting specific ARGs and provided comprehensive information about the resistome in the sample [36,40]. Moreover, identification of the reservoirs of ARGs carried by mobile genetic elements is difficult using the conventional metagenomic methods; however, the proximity ligation employed during the metagenomic Hi-C library preparation has made it possible to identify the hosts of such resistance genes. This novel method involves linking non-chromosomal DNA to the chromosome while the bacterial cell is still intact, which enables tracking the sources of the mobile genetic elements. Thus, the advantage of this method over other metagenomic approaches is that, besides detecting resistome in complex environmental samples, it can identify the bacterial host carrying the target resistance genes [40]. Given the power of this sequencing method to link specific genotypes to their taxonomic source, the objective of this study was to track bacterial reservoirs of ARGs conferring resistance to tetracyclines and other antibiotics in DD lesions using the metagenomic Hi-C method. The Hi-C approach holds promise for tracking the horizontal transfer of ARGs to other pathogens in complex microbial communities and their dissemination in the environments, including on animal farms.

## 2. Results

In this study, DD lesion samples from various disease stages were combined to capture an overall picture of antimicrobial resistance in DD. This mixture was used as an input to the ProxiMeta metagenomic Hi-C method to first reconstruct genomes of DD community members and then to examine the resistome profile of each such genome.

### 2.1. Microbiome Profiles

Metagenome deconvolution using ProxiMeta resulted in the creation of 308 putative genome and genome fragment clusters, with genome size ranging from 234,157 to 2,957,833 base pairs. However, only 40 of the clusters had ≥80% complete genome (Appendix A), 15.26% of the 308 clusters. These clusters were classified into five bacterial phyla (Table 1), namely, *Bacteroidetes* (relative abundance 5.27%), *Spirochaetes* (4.87%), *Firmicutes* (3.83%), *Tenericutes* (0.84), and *Proteobacteria* (0.45%). Moreover, some bacterial taxa were identified at the species level, including *Porphyromonas levii* (1.04%), *Treponema phagedenis* (1.0%), *Mycoplasma fermentans* (0.63%), *T. medium* (0.51%), *Porphyromonas somerae (0.36%),* and *Streptococcus henryi* (0.25%).

Out of the 40 clusters, 27 of them had an assembled 16S ribosomal RNA (rRNA) sequence with gene length ranging from 368 to 1555 base pairs. These 16S rRNA sequences were used to classify each such cluster more specifically. A best match organism, percent identity, and gene size of these bacterial clusters using 16S rRNA sequence BLAST (Basic Local Alignment Search Tool) are presented in Table 2. For instance, cluster 5 *Spirochaetaceae* had a 97.23% match with *Treponema refringens*, cluster 18 (*Spirochaetaceae)* a 99.01% match with *Treponema* species, cluster 13 (*Bacteroidetes)* an 89.96% match with *P. somerae*, and cluster 26 (*Porphyromonadaceae)* a 94.48% match with *P. levii*. More interestingly, six clusters only identified as kingdom *Bacteria* were further classified as *Spirochaetes* species (cluster 14), *P. levii* (cluster 25), *Acholeplasmatales* species (cluster 29), *Erysipelothrix rhusiopathiae* (cluster 32), *Bacteroidetes* species (cluster 36), and *Tenericutes* species (cluster 38).

#### Phylogenetic Tree

To study the genetic relatedness of *Spirochaetes*, *Bacteroidetes*, *Mycoplasma*, and unclassified bacterial clusters with frequently reported isolated bacterial species from DD cases, we plotted a phylogenetic tree (Figure 1). Some *Spirochaetes* clusters clustered with *T. denticola* ATCC 33520 and *Treponema pedis (*clusters 1, 2, 10, 15, and 28), while cluster 18 clustered around *T. medium* ATCC 700293 and *Treponema vincentii* F0403. Similarly, previously unreported *Bacteroidetes* and *Mycoplasma* genomes clustered around known and representative species, as indicated in Figure 1.

### 2.2. Resistome Profiles

Tetracycline resistance genes were identified by aligning assembled contigs to the curated resistance genes of the MEGARes database using minimap2. Hi-C data were then used to link identified resistance gene sequences to host genomes and genome fragments within ProxiMeta by counting the number of Hi-C reads linking such sequences to putative hosts. In this study, a total of 1961 tetracycline resistance genes, grouped into 16 classes, were identified (Table 3 and Appendix A). Genes encoding ribosomal protection proteins were predominant, with eight distinct classes, followed by resistance genes encoding efflux pump with six distinct classes, and one class encoding an inactivation enzyme. The most prevalent tetracycline resistance genes included *tet*Q (*n* = 1265), *tet*O (*n* = 224), *tet*W (*n* = 166), *tet*M (*n* = 112), *tet*40 (*n* = 41), *tet*32 (*n* = 39), and *tet*T (*n* = 32). Similarly, *tet*Q had the most taxonomic distribution with its presence detected in 21 of the 40 bacterial clusters with ≥80% complete genome (Figure 2), followed by *tet*O and *tet*M with each present in 11 clusters. However, some of the resistance genes, such as *tet*B, *tet*H, *tet*L, *tet*36, and *tet*S, were not identified in any of the bacterial taxa with ≥80% complete genomes.

Regarding tetracycline resistance gene reservoirs, all phyla identified in the present study hosted at least one type of tetracycline resistance gene. *Bacteroidetes, Firmicutes,* and *Spirochaetes* carried the three highest abundances of resistance genes with 445, 146, and 94 hits, respectively. In terms of diversity, *Firmicutes* hosted seven types of tetracycline resistance genes, *Bacteroidetes* six, *Spirochaetes* five, *Proteobacteria* three, and *Tenericutes* one. Additionally, all phyla hosted at least one class of the ribosomal protection protein genes, while efflux pump genes were hosted only by two phyla, including *Bacteroidetes* and *Proteobacteria*. The inactivation enzyme gene was identified only in *Firmicutes* (family *Clostridiales*). Classes and numbers of the tetracycline resistance genes with their respective bacterial hosts are depicted in Table 3.

Other ARGs that co-occurred with tetracycline determinants in the microbiome of DD lesions are summarized in Table 4. Resistant determinants against aminoglycoside (*n* = 463), beta-lactams (*n* = 36), sulfonamide (*n* = 39), phenicol (*n* = 20), lincosamide (*n* = 153), and erythromycin (*n* = 63) were detected in our samples. These resistance genes showed widespread distributions in the microbial community where all phyla, except *Tenericutes,* hosted multiple types of the ARGs. *Spirochaetes* was the lead host by abundance, with a total of 85 antimicrobial determinants that encoded resistance against aminoglycoside. In contrast, *Bacteroidetes* and *Proteobacteria* were hosts of more diverse resistance genes (four antimicrobials each).

## 3. Discussion

Lameness in cows caused by DD poses great challenges to dairy and beef farms across the world. As DD is a complex polymicrobial disease with many of the putative bacterial etiologies requiring fastidious growth conditions, the study of antimicrobial resistance associated with the disease is challenging. Published work characterizing the antimicrobial resistance status of the bacteria involved in the pathogenesis of DD is limited. This study, therefore, was designed to fill this research gap using the novel ProxiMeta Hi-C method that has the capability of identifying specific ARGs within their respective bacterial reservoirs directly from complex microbial communities. In the present study, we identified five phyla carrying multiple genes conferring resistance to tetracyclines and other antimicrobials.

*Bacteroidetes, Spirochaetes,* and *Firmicutes* were the most abundant phyla in the samples that we analyzed, which is in agreement with previous studies [14,19,21]. In this study, the lower-level classification of bacteria revealed the presence of *Pophyromonadaceae, Treponema, Clostridiales, Mycoplasma,* and *Lactobacillales,* which is also consistent with previous reports [5,14]. However, variations in the diversity and frequency of bacterial taxa among studies can be expected as the types of bacterial species involved in DD cases vary with the stage of the disease and the lesion sites [41]. For example, in the early stage of the lesions, *Bacteroides, Firmicutes (Peptostreptococcus, Peptococcus, Clostridium),* and *Fusobacterium* species are most abundant [42], while the later stages are dominated by *Treponema* species that appear in low relative abundance in the early stages [5,14,26]. Species of *Firmicutes* are the most significant and diverse bacteria associated with superficial and intermediate zones of the lesions, while *Treponema* species dominate the deeper layers of the lesions [5,26]. In the present study, samples were collected from representative lesions balanced for each stage (Iowa State University [ISU] DD Scores 1 to 4) at a slaughterhouse and macerated together.

Here, the 16S rRNA BLAST analysis and the phylogenetic analysis were used to identify and classify bacterial taxa at the kingdom, class, order, and family levels using the CheckM tool. For example, 6 of the 40 clusters were not classified beyond the kingdom (i.e., *Bacteria)*; however, sequence comparisons of their 16S rRNA sequences by BLAST showed that they have a close match with species of *Spirochaetes* and *Bacteroidetes*, along with *Erysiplothrix rhusiopathiae*. The involvement of the former two taxa in DD pathogenesis has been reported, while the role of the latter taxon is not yet known [5]. Similarly, one of the *Spirochaetes* clusters has a 97.23% identity match with *T. refringens*, which is in agreement with the findings of Nielsen and colleagues [18], where *T. refringens*-like species along with *T. phagedenis*-like species were the most abundant *Treponema* species in DD lesions. In the phylogenetic tree, the *Spirochaetes* clusters have shown some grouping patterns; clusters 1, 2, 10, 15, and 28 were clustered around *T. denticola* ATCC 33520, while cluster 18 was grouped around *T. medium* ATCC 700293 and *T. vincentii* F0403. In summary, the *Spirochaetes* species identified in this study were *T. phagedenis*, *T. medium*, *T. denticola-like species,* and *T. medium*-/*T. vincentii-like species,* which is consistent with previous studies [5,18,22,24].

We also found numerous tetracycline resistance genes in the DD samples. Typically, various treatment regimens of tetracyclines have been reported to result in a clinical improvement of DD in dairy cattle [29,43,44]. Previously, low efficacy of oxytetracycline treatment was observed in dairy cows with DD, which suggested possible antibiotic resistance development [30]. In the current study, we identified 16 classes of tetracycline resistance determinants encoding ribosomal protection proteins, efflux pumps, and enzymatic inactivation.

This study shows that tetracycline resistance genes are found in high abundance in the DD lesions, as demonstrated by the detection of large numbers of the core tetracycline resistome, such as *tetQ*, *tetO*, *tetM*, and *tetW*. Previously *tetA*, *tetM*, *tetO*, *tetQ*, *tetW*, *tetX*, and *tetY* were detected in cattle excrement, manure, and soil on dairy farms [45]. Compared with the current study, these genes, except for *tetA* and *tetY*, were detected in DD samples. Furthermore, we discovered two more gene types, tetL (four hits) and tetZ (six hits). All tetracycline resistance determinants detected here were reported previously from dairy farms and agricultural environments [45,46,47]. However, through the use of metagenomic Hi-C, we were able to report for the first time the presence of tetracycline resistance genes in a sample obtained from DD cases. This is particularly important for the specific fastidious bacteria involved in the pathogenesis of this disease. Since many of these bacteria grow poorly, or not at all, on artificial media, characterizing the emergence of antibiotic resistance among them using conventional methods is not possible [5]. However, using the Hi-C method enabled us to identify ARG hosting specific bacterial taxa [40].

In the present study, *tetQ* was the most prevalent and had the largest bacterial taxa distribution, in contrast to a previous study where *tetM* was the most abundant gene [47]. The discrepancy might be due to the difference in sample types; we used macerated biopsies of DD lesions, whereas Kobash et al. (2007) used swine and poultry feces. In general, this study demonstrates the widespread distribution of tetracycline resistance genes in the microbiota of DD lesions.

The ribosomal protection protein-encoding genes were also found to be most dominant, agreeing with a previous report [48]. However, some studies documented efflux pump genes as dominant [46,49]. Previously, 28 distinct classes of genes encoding the efflux pumps and 12 distinct classes encoding ribosome protection proteins were reported [46]. Some of the ribosomal protection protein-encoding genes, such as *tetM* and *tetW*, are associated with conjugative transposons and their host range has increased dramatically in recent years [46]. Besides that, similar to our findings, *tetX* and *tet*37 that encode less prevalent resistance mechanisms such as monooxygenases and mutations within the 16S rRNA were reported [49]. In the present study, we detected *tet*X in *Firmicutes*. On the other hand, two of the identified bacterial taxa were found to carry *tet*31, and one bacterial taxon carried *tet*33, also consistent with a previous study [46]. *tet*31 and *tet*33 confer resistance by inducing the efflux pump mechanism.

*Bacteroidetes* species were found to host the largest number of tetracycline resistance genes (*n* = 445) compared to other phyla with similar relative abundance. Except for one gene (encoding for the efflux pump), the rest of these genes were ribosomal protection protein-encoding genes. The increase in the carriage of tetracycline resistance genes, such as tetQ, by this phylum, was previously noted [50,51,52]. In agreement with the present study, *tet*M and *tet*W was isolated from *Porphyromonadaceae*, a family in this phylum [46]. Horizontal transfer of tetracycline resistance genes among *Bacteroides* species and between *Bacteroides* species that colonize humans and livestock were also documented [53,54]. Conjugative transposons and plasmids play a significant role in the horizontal transfer of these resistance genes among *Bacteroides* species and in the increase of tetracycline resistance among them [54,55]. Although antimicrobial resistance studies related to livestock isolates are absent, studies in human isolates suggest that horizontal transfer of resistance genes mainly contributes to the high carriage of ARGs in *Bacteroidetes*.

*Firmicutes* hosted 146 tetracycline resistance genes classified into seven types of genes that, except for one, encoded ribosomal protection genes. *Firmicutes* was unique from other phyla in that it was the only phylum that carried the *tet*X gene encoding the inactivation enzyme. This gene confers resistance to all clinically important tetracycline drugs, including tigecycline, a broad-spectrum and last-resort antibiotic for multidrug-resistant pathogens [56]. Among the three bacterial taxa identified in *Firmicutes*, *Clostridiales* carried 143 tetracycline resistance genes. Like the present study, *tet*M was reported from *Lactobacillales* and *Clostridiales* from dairy farms in the Czech Republic [45], *tetW* from *Clostridiales* and *Lactobacillales*, and *tet*32 from *Clostridiales* [46]. *Firmicutes* species are among the most abundant bacterial taxa in DD lesions [14,19].

*Spirochaetes* were found to carry 94 tetracycline resistance genes, representing 5 classes of ribosomal protection genes. Spirochete treponemes are the most predominant microorganisms identified in DD lesions of cattle [5,19,20,24]. However, studies of antimicrobial resistance status of *Treponema* isolates are lacking, particularly for DD cases. Human isolates of *Treponema* species, such as *T. denticola* [57] and *Treponema pallidum* [58], have been found to carry tetracycline resistance genes. *tetB* was detected in oral *Treponema* isolates obtained from a human periodontal patient [57]. Significant genetic relatedness between oral *Treponema* isolates from humans and *Treponema* isolates from DD in cattle has been reported [5].

*Tenericutes* was found to host only one tetracycline resistance gene, *tetM*, which encodes ribosomal protection proteins. There is no report on the tetracycline resistance status of *Mycoplasma* isolates from DD lesions. However, other tetracycline-resistant *Mycoplasma* isolates from cattle and humans have been described [59,60,61]. The emergence of tetracycline resistance in *Mycoplasma* species was attributed to *tetM* acquisition [59].

*Proteobacteria* hosted three *tet*31 genes encoding the efflux pump and one tetW gene encoding ribosomal protection proteins. The role of *Proteobacteria* species in the pathogenesis of DD is not well understood, but they are more abundant in DD lesions than healthy skin [9,19]. However, if these antimicrobial determinants are carried by mobile genetic elements such as plasmids, *Proteobacteria* species might transfer these genes to other pathogenic bacteria, even if they play no role in the disease process.

We also observed the co-occurrence of other antimicrobial resistance genes conferring resistance to aminoglycosides, beta-lactams, sulfonamides, phenicol, lincosamides, and erythromycins in the sample. Some of these antimicrobials are used for the treatment of DD: penicillin (beta-lactams) systematically, lincomycin/spectinomycin (lincosamide) and chloramphenicol (phenicol in the UK, but illegal in the USA) topically, and erythromycin and lincomycin/spectinomycin in foot baths [62]. The rest of them are used for the treatment of different diseases on dairy farms [63,64]. The detection of resistance genes against these antimicrobials in the DD lesions has not been reported thus far; however, their existence in samples from livestock, livestock products, and farm environments has been documented [65,66,67]. Similarly, the antimicrobial susceptibility test of *Treponema phagedenis*-like spirochetes isolated from dairy cattle with DD lesions in Japan showed that oxytetracycline, lincomycin, enrofloxacin, chloramphenicol, ceftiofur, and gentamicin had intermediate MIC values [31]. In agreement with our findings, genes conferring resistance to multiple antimicrobials such as tetracycline, sulfonamide, quinolone, macrolides (erythromycin), and aminoglycoside were detected in wastewater and surface water samples collected from pig, cattle, and chicken farms in China (Chen et al., 2015). These resistance genes and their bacterial reservoirs can pollute the environment and jeopardize public health [68,69]. This warrants concerted efforts among human, animal, and environmental health key players to slow down the emergence of antimicrobial resistance in general and to mitigate the dissemination of the resistance genes in the environment, in particular. Genes conferring resistance to commonly used footbath heavy metals, such as copper and zinc sulfate, were not detected in this study, which is in contrast to a previous report [19].

## 4. Materials and Methods

### 4.1. Description of the Sample

The sample used in this study was obtained from large pooled samples collected from cows with lesions of DD at slaughterhouses for the purpose of inducing lesions experimentally [70]. The skin lesions were biopsied from cows with DD at various stages. A scoring system that is based on the morphologies of DD lesions and representing various stages of DD progression was previously developed in our lab [14]. Biopsy samples were collected from DD lesions representative of each score. The harvested biopsy materials were placed into Induction Broth, and then combined and macerated in an anaerobic chamber as described in our publication [70]. The macerated samples were stored at −80 °C in a freezer. For this study, an aliquot of inoculum was thawed at room temperature for DNA extraction and laboratory preparation according to the protocols of Phase Genomics and ZymoBiomics.

### 4.2. DNA Extraction and Library Preparation

#### 4.2.1. DNA Extraction for Shotgun Metagenomics

DNA was extracted from the sample according to ZymoBIOMICS instructions. Briefly, the sample was thawed at room temperature for 30 min. A 100 mg sample was transferred to a 2 mL ZR BashingBead lysis tube and mixed with 250 µL deionized sterile water, 750 µL lysis solution, and 50 µL proteinase K. The samples were processed by a bead beater for 10 min followed by incubation for at least 30 min in a water bath at 55 °C. Then, the lysis tubes were centrifuged in a microcentrifuge at 10,000× *g* for 3 min. The supernatant was harvested to columns and then washed with DNA Wash Buffer 1 and 2. The final product was eluted with 75 µL DNase/RNase-free water. The concentration of eluted DNA was measured first by NanoDrop 3300 Fluorospectrometer and confirmed by the Qubit Fluorometer. The whole-genome extracts were submitted to the DNA Facility of Iowa State University, where a single flow cell lane Illumina HiSeq platform was used for sequencing (2 × 150 base pairs).

#### 4.2.2. DNA Extraction and Library Preparation for ProxiMeta Hi-C Metagenomics

We used a Phase Genomics (Seattle, WA, USA) ProxiMeta Hi-C Microbiome Kit, which is a commercially available version of the Hi-C protocol, for DNA extraction and library preparation. Briefly, 100 mg of the same sample used in the shotgun procedures was washed with Tris-buffered saline (TBS), followed by in vivo crosslinking of genetic materials (both chromosomal and non-chromosomal) using a formaldehyde solution, while the bacterial cells were still intact. Subsequently, it was digested using the Sau3AI and MlucI restriction enzymes and proximity ligated with biotinylated nucleotides to create chimeric molecules composed of fragments from different regions of genomes that were physically proximal in vivo according to the manufacturer’s instructions for the kit. The chance of inter-cellular interactions of genetic materials is negligible. Following the protocol, molecules were pulled down with streptavidin beads and processed into an Illumina-compatible sequencing library. Sequencing was performed on an Illumina HiSeq instrument (2 × 150 base pairs) at the DNA Facility of Iowa State University.

#### 4.2.3. Metagenomic Data Analysis

Shotgun and Hi-C metagenomic sequencing files were uploaded to the Phase Genomics cloud-based bioinformatics portal for subsequent analysis. Shotgun reads were filtered and trimmed for quality using fastp [71] and then assembled with MEGAHIT [72,73] using default options. Following the instructions of the kit, Hi-C reads were aligned to the assembly (https://phasegenomics.github.io/2019/09/19/hic-alignment-and-qc.html (accessed on 22 February 2021)). Briefly, reads were aligned using BWA-MEM [74] with the −5SP and −t 8 options specified, and all other options default. SAMBLASTER [75] was used to flag PCR duplicates, which were later excluded from the analysis. Alignments were then filtered with samtools [76] using the −F 2304 filtering flag to remove non-primary and secondary alignments. Metagenome deconvolution was performed with ProxiMeta [37,38]. Clusters were assessed for quality using CheckM [77] and assigned preliminary taxonomic classifications with Mash [78]. Identification of ARGs was conducted by aligning assembled contigs to curated ARGs of the MEGARes database using minimap2 [79]. Hi-C data were then used to link identified ARG sequences to host genomes and genome fragments within ProxiMeta by counting the number of Hi-C reads linking such sequences to putative hosts [36,40].

### 4.3. 16S rRNA BLASTing and Phylogenetic Tree Plotting

From the metagenomics, we obtained 40 bacterial clusters (taxa) with ≥80% complete genome. The clusters were annotated using the PATRIC online tool (www.patricbrc.org (accessed on 22 February 2021)) to search for the best matching bacterial species for those clusters whose 16S rRNA gene was assembled from the metagenome data. The 16S rRNA sequence was blasted on the website of the U.S. National Center for Biotechnology Information (https://www.ncbi.nlm.nih.gov/ (accessed on 22 February 2021)). Similarly, PATRIC online tool was used to plot the phylogenetic trees of the clusters; previously reported bacterial isolates from DD and other references had been used as reference organisms on the dendrogram.

## 5. Conclusions

The inclusion of a DNA crosslinking step before DNA extraction and subsequent proximity ligation in the Hi-C metagenomics enabled individual microbial genomes to be deconvoluted from within metagenomic DNA samples. Thus, antimicrobial resistance genes were identified and assigned to their taxonomic sources. Our study shows that tetracycline resistance genes are widely distributed in bacteria that are believed to be involved in the pathogenesis of DD. Similarly, we identified the co-occurrence of resistance determinants to other commonly used antimicrobials with their respective bacterial hosts, which demonstrates the superiority of the Hi-C approach over other metagenomic methods in tracking ARGs in a complex microbial community. To our knowledge, this is the first study that provides genetic evidence of resistance to tetracycline and other antimicrobials harbored by bacteria that are involved in the pathogenesis of DD. The clinical importance of these findings is doubtless, but its interpretation requires precaution. Furthermore, the high abundance and diversity of tetracycline resistance genes, the most common antibiotic used for the treatment of DD, in the sequenced sample may provide insights into the incomplete resolution of many DD lesions following aggressive treatment with tetracyclines. Future efforts will use the Hi-C metagenomic approach to substantiate the role of horizontal gene transfer in the dissemination of antimicrobial determinants on farms as well as environments.

## Figures and Tables

**Figure 1 antibiotics-10-00221-f001:**
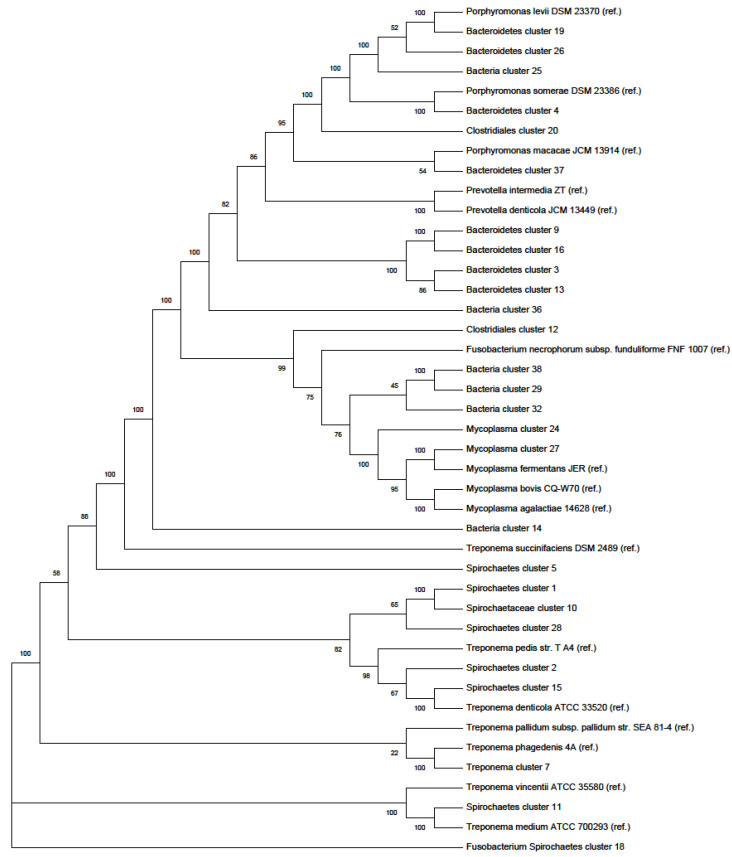
Phylogenetic tree of *Spirochaetes*, *Bacteroidetes*, *Mycoplasma*, and *Bacteria* from this study and reference organisms (which are labeled as “ref.”). In the construction of this dendrogram, we included the sequences of selected organisms identified in the current study and previous studies as well reference organisms, and PATRIC online tool (https://www.patricbrc.org/app/PhylogeneticTree (accessed on 22 February 2021)) was used.

**Figure 2 antibiotics-10-00221-f002:**
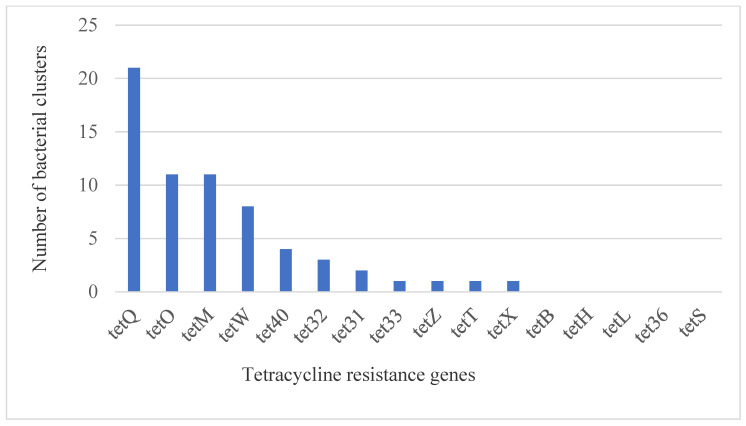
Numbers of bacterial clusters with ≥80% complete genome coverage carrying tetracycline resistance genes.

**Table 1 antibiotics-10-00221-t001:** Relative abundances of bacterial taxa with ≥80% complete genome identified in digital dermatitis lesions. These bacterial taxa account for 15.26% of the total bacterial taxa detected in the sample.

Phyla	Lowest Level Identification	Classification	Relative Abundance (%)
***Bacteroidetes***		**Phylum**	**5.27**
	*Porphyromonas somerae*	Species	0.36
	*Porphyromonas levii*	Species	1.04
	Porphyromonadaceae (unclassified)	Family	0.88
	*Bacteroidetes* (unclassified)	Phylum	2.99
***Spirochaetes***		**Phylum**	**4.87**
	*Treponema phagedenis*	Species	1.00
	*Treponema medium*	Species	0.51
	*Spirochaetaceae* (unclassified)	Family	3.36
***Firmicutes***		**Phylum**	**3.83**
	*Streptococcus henryi*	Species	0.25
	*Lactobacillales* (unclassified)	Order	1.15
	*Clostridiales* (unclassified)	Order	2.43
***Tenericutes***		**Phylum**	**0.84**
	*Mycoplasma fermentans*	Species	0.63
	*Mycoplasma* (unclassified)	Genus	0.21
***Proteobacteria***		**Phylum**	**0.45**
	*Betaproteobacteria* (unclassified)	Class	0.26
	*Gammaproteobacteria* (unclassified)	Class	0.19

**Table 2 antibiotics-10-00221-t002:** Best match bacterial species, percent identity, and gene sizes of 27 bacterial clusters using 16S ribosomal RNA sequence blasting on the National Center for Biotechnology Information (NCBI) website.

Cluster-ID	Top References	Best Match	Identity (%)	16S rRNA Length (Bases)
cluster_3	*p_Bacteroidetes*	Uncultured bacterium clone gls269	84.42	1511
cluster_4	*Porphyromonas_somerae_DSM_23386*	*Porphyromonas somerae* DSM 23386 strain JCM 13867	98.35	567
cluster_5	*f_Spirochaetaceae*	*Treponema* *refringens*	97.23	1365
cluster_6	*p_Bacteroidetes*	*Bacteroidia bacterium* feline oral taxon 312 clone UI046	93.38	707
cluster_8	*c_Betaproteobacteria*	*Oligella ureolytica* DSM 18253	96.51	1540
cluster_12	*o_Clostridiales*	*Ezakiella sp. Marseille*-P2951 strain Marseille-P2951T	99.73	747
cluster_13	*p_Bacteroidetes*	*Porphyromonas somerae* DSM 23386 strain JCM 13867	89.96	1113
cluster_14	*k_Bacteria*	*Spirochaeta sp. canine* oral taxon 314 clone 1A090	99.92	1234
cluster_16	*p_Bacteroidetes*	*Bacteroidia bacterium feline* oral taxon 312 clone UI046	85.81	721
cluster_17	*o_Clostridiales*	*Catonella* sp. oral clone BR063	94.1	874
cluster_18	*f_Spirochaetaceae*	*Treponema sp. canine* oral taxon 233 clone QB043	99.01	414
cluster_19	*Porphyromonas_levii_DSM_23370*	*Porphyromonas levii* strain Israel	99.4	368
cluster_21	*o_Clostridiales*	Uncultured rumen bacterium	98.4	566
cluster_23	*Streptococcus_henryi_DSM_19005*	*Streptococcus henryi* strain OZK31	98.45	916
cluster_24	*g_Mycoplasma*	*Mycoplasma agalactiae* 5632	93.52	802
cluster_25	*k_Bacteria*	*Porphyromonas levii* DSM 23370 strain JCM 13866	94.59	398
cluster_26	*f_Porphyromonadaceae*	*Porphyromonas levii* DSM 23370 strain JCM 13866	94.48	372
cluster_27	*Mycoplasma_fermentans_JER*	*Mycoplasma fermentans* M64	99.87	1522
cluster_29	*k_Bacteria*	*Acholeplasmatales bacterium* canine oral taxon 172 clone QC046	98.6	1555
cluster_30	*c_Gammaproteobacteria*	*Pseudomonas sp. M-08* gene	97.87	1324
cluster_31	*o_Clostridiales*	Uncultured bacterium clone 1101352040638	93.16	1536
cluster_32	*k_Bacteria*	*Erysipelothrix rhusiopathiae*	92.17	611
cluster_33	*o_Clostridiales*	*Peptoniphilaceae* bacterium SIT14	96.86	1536
cluster_35	*o_Clostridiales*	Uncultured *Firmicutes* bacterium clone P-07	99.68	1510
cluster_36	*k_Bacteria*	Uncultured *Bacteroidetes* bacterium clone BL2_5	98.27	1536
cluster_38	*k_Bacteria*	Uncultured *Tenericutes* bacterium clone P-06	100	1542
cluster_39	*o_Clostridiales*	*Clostridium sticklandii* str. DSM 519	85.56	471

**Table 3 antibiotics-10-00221-t003:** Types and distributions of tetracycline resistance genes in bacterial taxa having ≥80% complete genome detected in digital dermatitis lesions.

Phyla	Bacterial Taxa	Efflux Pump						Ribosomal Protection Proteins									Inactivation Enzyme
		*tet*31	*tet*33	*tet*B	*tet*H	*tet*L	*tet*Z	*tet*32	*tet*36	*tet*40	*tet*M	*tet*O	*tet*O	*tet*S	*tet*T	*tet*W	*tet*X
***Bacteroidetes***		**0**	**0**	**0**	**0**	**0**	**1**	**0**	**0**	**1**	**3**	**3**	**435**	**0**	**0**	**2**	**0**
	*Porphyromonas somerae*	0	0	0	0	0	0	0	0	0	2	0	19	0	0	0	0
	*Porphyromonas levii*	0	0	0	0	0	0	0	0	0	0	0	86	0	0	1	0
	Porphyromonadaceae (unclass. *)	0	0	0	0	0	0	0	0	1	1	1	68	0	0	0	0
	*Bacteroidetes* (unclass.)	0	0	0	0	0	1	0	0	0	0	2	262	0	0	1	0
***Spirochaetes***		**0**	**0**	**0**	**0**	**0**	**0**	**1**	**0**	**0**	**23**	**64**	**5**	**0**	**1**	**0**	**0**
	*Treponema phagedenis*	0	0	0	0	0	0	0	0	0	2	63	1	0	0	0	0
	*Treponema medium*	0	0	0	0	0	0	0	0	0	14	0	0	0	0	0	0
	*Spirochaetaceae* (unclass.)	0	0	0	0	0	0	1	0	0	7	1	4	0	1	0	0
***Firmicutes***		**0**	**0**	**0**	**0**	**0**	**0**	**32**	**0**	**7**	**4**	**71**	**12**	**0**	**0**	**20**	**1**
	*Streptococcus henryi*	0	0	0	0	0	0	0	0	0	1	0	0	0	0	1	0
	*Lactobacillales* (unclass.)	0	0	0	0	0	0	0	0	0	1	0	0	0	0	0	0
	*Clostridiales* (unclass.)	0	0	0	0	0	0	32	0	7	2	71	12	0	0	19	1
***Tenericutes***		**0**	**0**	**0**	**0**	**0**	**0**	**0**	**0**	**0**	**1**	**0**	**0**	**0**	**0**	**0**	**0**
	*Mycoplasma fermentans*	0	0	0	0	0	0	0	0	0	1	0	0	0	0	0	0
	*Mycoplasma* (unclass.)	0	0	0	0	0	0	0	0	0	0	0	0	0	0	0	0
***Proteobacteria***		**3**	**0**	**0**	**0**	**0**	**0**	**0**	**0**	**0**	**0**	**0**	**6**	**0**	**0**	**1**	**0**
	*Betaproteobacteria* (unclass.)	2	0	0	0	0	0	0	0	0	0	0	0	0	0	0	0
	*Gammaproteobacteria* (unclass.)	1	0	0	0	0	0	0	0	0	0	0	6	0	0	1	0
**Total Resistance Genes in the** **Sample (*n* = 308 clusters) ****																	
		**13**	**4**	**1**	**22**	**4**	**6**	**39**	**14**	**41**	**112**	**224**	**1265**	**15**	**32**	**166**	**3**

* Not further classified. ** This row provides the total numbers of antimicrobial resistance genes (ARGs) in the sample; thus, the sum of a specific ARG in bacterial clusters with ≥80% complete genomes does not sum up to the total number of that specific ARG or column.

**Table 4 antibiotics-10-00221-t004:** Aminoglycoside, beta-lactam, sulfonamide, phenicol, lincosamide, and erythromycin resistance determinants hosted by microbiota of digital dermatitis lesions for 40 clusters with ≥80% complete genome; resistance genes are in parentheses.

Phyla	Bacterial Taxa	Aminoglycoside (*aad*A1, *ant*3, *ant*4, *ant*6, *aph*3, aph6)	Beta-Lactam (*bla*OXA, *bla*CARB)	Sulfonamide (*sul*1, *sul*2)	Phenicol (*cmx*AB, *flo*R)	Lincosamide (*lnu*B, *lsa*)	Erythromycin (*erm*A, *erm*B, *erm*F, *erm*X, *myr*A)
***Bacteroidetes***		**8**	**0**	**3**	**0**	**1**	**4**
	*Porphyromonas somerae*	0	0	3	0	0	0
	*Porphyromonas levii*	3	0	0	0	0	1
	*Porphyromonadaceae*	0	0	0	0	1	1
	*Bacteroidetes*	5	0	0	0	0	2
***Spirochaetes***		**85**	**0**	**0**	**0**	**0**	**0**
	*Treponema phagedenis*	1	0	0	0	0	0
	*Treponema medium*	9	0	0	0	0	0
	*Spirochaetaceae*	75	0	0	0	0	0
***Firmicutes***		**9**	**0**	**0**	**0**	**28**	**2**
	*Streptococcus henryi*	3	0	0	0	10	0
	*Lactobacillales*	1	0	0	0	5	0
	*Clostridiales*	5	0	0	0	13	2
***Tenericutes***		**0**	**0**	**0**	**0**	**0**	**0**
	*Mycoplasma fermentans*	0	0	0	0	0	0
	*Mycoplasma*	0	0	0	0	0	0
***Proteobacteria***		**11**	**1**	**1**	**1**	**0**	**0**
	*Betaproteobacteria*	6	1	1	0	0	0
	*Gammaproteobacteria*	5	0	0	1	0	0
**Total Resistance Genes in the Sample (n = 308 clusters)**		**463**	**36**	**39**	**20**	**153**	**63**

## Data Availability

Metagenomic shotgun and Hi-C sequence data are available at SRA accession PRJNA704056. Processed data for linking contigs to genome clusters using Hi-C data are available at https://osf.io/4wnh2/ (accessed on 22 February 2021).

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
