# Peer review of "Tracking Reservoirs of Antimicrobial Resistance Genes in a Complex Microbial Community Using Metagenomic Hi-C: The Case of Bovine Digital Dermatitis"

_antibiotics, 2021, doi:10.3390/antibiotics10020221_

Round 1

Reviewer 1 Report

This manuscript by Beyi et al. entitled “Tracking reservoirs of antimicrobial resistance genes in a complex microbial community using metagenomic Hi-C: the case of bovine digital dermatitis” is an interesting, well-written manuscript. The authors clearly describe multiple important problems as the basis for the paper: 1) the occurrence and cost of bovine digital dermatitis (DD), 2) the lack of knowledge surrounding the causative agents of DD due to difficulties in culturing these microbes, and 3) potential spread of antibiotic resistance genes (ARGs) due to extended usage in agriculture, with the focus on tetracycline resistance genes. The authors take an interesting approach to metagenomics by using a Hi-C based approach to perform proximity-dependent ligation of intracellular genomic material, which enables identification of ARGs and also determination of which genes are encoded by the same organism as other ARGs.

Overall, the introduction is well written and provides all of the necessary background information with appropriate citations. The ARG data is presented clearly. The major concern I have is that the sequences (raw files and clusters with 80% complete genome) are not deposited in SRA or ENA, making it impossible to reproduce the results of this study or do any addition computational analysis of their datasets.

Additional minor comments:

Lines 86-87: Given that Hi-C based approaches are relatively uncommon in bacterial metagenomics (certainly there is less familiarity with the approach as opposed to 16S sequencing or other metagenomics approaches), it would be helpful to expand the discussion of how Hi-C differs from other metagenomic approaches and the benefits of using Hi-C over other metagenome deconvolution approaches.

Line 113: “, and” should not be italicized

Line 140, legend for Figure 1: Additional information such as input sequence and methods used for tree building are needed.

Discussion section: Why are ribosomal protection genes so much more widespread than inactivation enzymes/efflux pumps? Is this typically observed in metagenomic studies of ARGs from agricultural reservoirs?

Lines 203-204: This scoring section is mentioned here but not in the Methods section. Please elaborate on any scoring/sample identification that was used. Do the authors think that combining samples from multiple stages affected the overall results of the study?

Line 204: Is the maceration approach harsh enough to break open cells, and if so, does that complicate the Hi-C method and interpretation of data?

Discussion section: A large portion of the discussion section (including parts of the paragraphs discussion distribution of tetR genes) seems as though it would be better placed in the results section.

Author Response

Dear reviewer,

We greatly appreciate your comments and suggestions that you provided on our MS. We considered all of them valuable; and thus, we have made changes based on them in the MS and provided answers to your questions in this cover letter; please kindly find them below.

Overall, the introduction is well written and provides all of the necessary background information with appropriate citations. The ARG data is presented clearly. The major concern I have is that the sequences (raw files and clusters with 80% complete genome) are not deposited in SRA or ENA, making it impossible to reproduce the results of this study or do any addition computational analysis of their datasets.

Thank you for your suggestions and comments on our work. With regard to uploading the clusters with 80% complete genomes (i.e., 40 clusters), the submission process imposes a set of filters on the assemblies such that they may change the clusters (for example, they get rid of contigs below some length threshold). This problem happened in a previous effort. For this reason, we would like to attach the clusters to the paper as a supplemental data archive, to ensure that they are exactly the same if this journal provides that platform. We are working on depositing the raw data in SRA.

Additional minor comments:

Lines 86-87: Given that Hi-C based approaches are relatively uncommon in bacterial metagenomics (certainly there is less familiarity with the approach as opposed to 16S sequencing or other metagenomics approaches), it would be helpful to expand the discussion of how Hi-C differs from other metagenomic approaches and the benefits of using Hi-C over other metagenome deconvolution approaches.

We agree that metagenomic Hi-C is a new method that requires more elaboration, especially about its advantages over other metagenomic methods. One of its advantage is identifying the sources of mobile genetic elements; it is an established fact that most ARGs are carried and disseminated by mobile genetic elements. Hi-C involves crosslinking chromosomal and non-chromosomal DNA togethers while the bacterial cell is still intact, which enables tracking the reservoirs of non-chromosomal ARGs. These additional points have been noted in the revised version in lines 94 to 101. We greatly appreciate your suggestion.

Line 113: “, and” should not be italicized. It is corrected.

Line 140, legend for Figure 1: Additional information such as input sequence and methods used for tree building are needed.

We appreciate your suggestion on Figure 1. In the revised MS, we have added additional information that describes input sequences and an online tool used for the construction of dendrogram.

Discussion section: Why are ribosomal protection genes so much more widespread than inactivation enzymes/efflux pumps? Is this typically observed in metagenomic studies of ARGs from agricultural reservoirs?

We are grateful for these excellent questions. It is not possible to answer these questions based on the present study but few points can be made based on literature.

  • A review by Roberts (2005, cited in this MS) shows that the host range of tetM and tetW increased from 24 to 42 and 2 to 19 genera over a few years, which was attributed to their association with conjugative transposons that also have a very wide host range. tetW and tetM are ribosomal protection protein encoding genes and the third and fourth most abundant ARG in this study. This point is included in the revised submission.
  • Obviously, microbial diversity, and subsequently the resistome, of a farm environment is different from hospital or other environments. That means the types and distributions of ARGs are also different. A few metagenomic studies on farm or farm environmental samples indicate that the ribosomal protection-proteins encoding resistance genes are dominant. However, to make a solid conclusion a systematic review of available literature is required.

Lines 203-204: This scoring section is mentioned here but not in the Methods section. Please elaborate on any scoring/sample identification that was used. Do the authors think that combining samples from multiple stages affected the overall results of the study?

We appreciate your suggestions and question. Yes, previous studies show that the species of bacteria identified in DD lesions depends on the stage and the site of the lesions from where biopsies were taken from. Similarly, diversities of microbiota, and most likely the resistome, vary along with the stage and the site of the lesions. We tried to explain these phenomena in the discussion section, lines 212-219. More descriptions about the scoring system and a citation have been also provided in this revision, lines 339-345.

Line 204: Is the maceration approach harsh enough to break open cells, and if so, does that complicate the Hi-C method and interpretation of data?

We are grateful for your concern and question. The maceration approach was not strong enough to cause lysis of bacterial cells. After harvesting biopsy materials from cows with naturally DD lesions, the samples were placed in Induction Broth and macerated in an anaerobic chamber, which was not capable of breaking microbial cells. Now, we have included this additional description of sample in the Materials and Methods section lines 339 to 345.

Discussion section: A large portion of the discussion section (including parts of the paragraphs discussion distribution of tetR genes) seems as though it would be better placed in the results section.

Thank you for the suggestions. We agree that some of the sentences/paragraphs were redundant and some of them would be better placed in the Results section. We have made the following changes in this revision:

Removed

Lines 239-240: “These genes included tetQ (n = 1,265), tetO (n = 224), tetW (n = 166), tetM (n = 112), tet40 (n = 41), tet32 (n= 39), and tetT (n = 32).”

Moved to the Results section

Lines 255-258: “All phyla identified in the present study hosted at least one type of tetracycline resistance gene. Bacteroidetes, Firmicutes, and Spirochaetes carried the three highest abundances of resistance genes with 445, 146, and 94 hits, respectively. In terms of diversity, Firmicutes hosted seven types of tetracycline resistance genes, Bacteroidetes six, Spirochaetes five, Proteobacteria three, and Tenericutes one.”

Please kindly find the changes in the attached revised paper.

Reviewer 2 Report

Ashenafi et al. showed the metagenomics dates of bovine DD.

Almost nothing to information about the samples.

Authors need to write the method about analysis of metagenomics dates.

Table 3 is not adequate (non-match the number (total and each number)

Author Response

Dear reviewer,

We greatly appreciate your comments and suggestions that you made on this MS. We considered all of them valuable and we have made changes based on them in the revised submission and provided explanation to your suggestions in this cover letter; please kindly find them below.

Ashenafi et al. showed the metagenomics dates of bovine DD.

 Almost nothing to information about the samples.

We greatly appreciate your feedback. We agree that only a brief description of the sample used in this study was provided in our MS. Now we have added more information that explains how the samples were collected and macerated in the Materials and methods section lines 339 and 343. Our publication that provides detail description of these samples is also cited in this revised MS.

Authors need to write the method about analysis of metagenomics dates.

Thank you for your comment on metagenomic data analysis. We added a subtitle to this subsection to delineate it from lab procedures in this revision. Main steps have been mentioned in this subsection to elaborate the Hi-C and metagenomic data analyses; additionally, an extensive citation has been made to support our approach with literature. We believe that the provided description combined with citations will make this subsection reproducible.

Table 3 is not adequate (non-match the number (total and each number)

We are grateful for your comments. We understand that additional information is required to clarify this table. As described in the result section, in this study 308 bacterial clusters were identified, of which 40 of them had ≥80% complete genomes. In this table, we summarized ARGs in the 40 clusters as well as in 308 clusters, which was presented in the last row. However, to clarify this table we added the following information as a footnote. “This row provides the total numbers of ARGs in the sample; thus, the sum of a specific ARG in bacterial clusters with ≥80% complete genomes do not sum up to the total number of that specific ARG or column.”

Please kindly find the changes in the attached revised paper.

Round 2

Reviewer 2 Report

Thanks for revision.